# PREDICT-THEN-OPTIMIZE VIA LEARNING TO OPTIMIZE FROM FEATURES

## ABSTRACT

The Predict-Then-Optimize framework uses machine learning models to predict unknown parameters of an optimization problem from features before solving. Recent works show that decision quality can be improved in this setting by solving and differentiating the optimization problem in the training loop, enabling end-to-end training with loss functions defined directly on the resulting decisions. However, this approach can be inefficient and requires handcrafted, problem-specific rules for backpropagation through the optimization step. This paper proposes an alternative method, in which optimal solutions are learned directly from the observable features by predictive models. The approach is generic, and based on an adaptation of the Learning-to-Optimize paradigm, from which a rich variety of existing techniques can be employed. Experimental evaluations show the ability of several Learning-to-Optimize methods to provide efficient, accurate, and flexible solutions to an array of challenging Predict-Then-Optimize problems.

## 1 INTRODUCTION

The *Predict-Then-Optimize* (PtO) framework models decision-making processes as optimization problems whose parameters are only partially known while the remaining, unknown, parameters must be estimated by a machine learning (ML) model. The predicted parameters complete the specification of an optimization problem which is then solved to produce a final decision. The problem is posed as estimating the solution $\boldsymbol{x}^{\star}(\zeta) \in \mathcal{X} \subseteq \mathbb{R}^n$ of a *parametric* optimization problem:

$$\boldsymbol{x}^{\star}(\boldsymbol{\zeta}) = \arg\min_{\boldsymbol{x}} \ f(\boldsymbol{x}, \boldsymbol{\zeta}) \tag{1}$$

$$\text{such that: } \boldsymbol{g}(\boldsymbol{x}) \leq 0, \ \ \boldsymbol{h}(\boldsymbol{x}) = 0,$$

given that parameters $\boldsymbol{\zeta} \in \mathcal{C} \subseteq \mathbb{R}^p$ are unknown, but that a correlated set of observable values $\boldsymbol{z} \in \mathcal{Z}$ are available. Here $f$ is an objective function, and $\boldsymbol{g}$ and $\boldsymbol{h}$ define the set of the problem's inequality and equality constraints. The combined prediction and optimization model is evaluated on the basis of the optimality of its downstream decisions, with respect to $f$ under its ground-truth problem parameters (Elmachtoub & Grigas, 2021). This setting is ubiquitous to many real-world applications confronting the task of decision-making under uncertainty, such as planning the shortest route in a city, determining optimal power generation schedules, or managing investment portfolios. For example, a vehicle routing system may aim to minimize a rider's total commute time by solving a shortest-path optimization model (1) given knowledge of the transit times $\boldsymbol{\zeta}$ over each individual city block. In absence of that knowledge, it may be estimated by models trained to predict local transit times based on exogenous data $\boldsymbol{z}$, such as weather and traffic conditions. In this context, more accurately predicted transit times $\hat{\boldsymbol{\zeta}}$ tend to produce routing plans $\boldsymbol{x}^{\star}(\hat{\boldsymbol{\zeta}})$ with shorter overall commutes, with respect to the true city-block transit times $\boldsymbol{\zeta}$.

However, direct training of predictions from observable features to problem parameters tends to generalize poorly with respect to the ground-truth optimality achieved by a subsequent decision model (Mandi et al., 2023; Kotary et al., 2021b). To address this challenge, *End-to-end Predict-Then-Optimize* (EPO) (Elmachtoub & Grigas, 2021) has emerged as a transformative paradigm in data-driven decision making in which predictive models are trained by directly minimizing loss functions defined on the downstream optimal solutions $\boldsymbol{x}^{\star}(\hat{\boldsymbol{\zeta}})$.

On the other hand, EPO implementations require backpropagation through the solution of the optimization problem (1) as a function of its parameters for end-to-end training. The required back-

propagation rules are highly dependent on the form of the optimization model and are typically derived by hand analytically for limited classes of models (Amos & Kolter, 2017; Agrawal et al., 2019a). Furthermore, difficult decision models involving nonconvex or discrete optimization may not admit well-defined backpropagation rules.

To address these challenges, this paper outlines a framework for training Predict-Then-Optimize models by techniques adapted from a separate but related area of work that combines constrained optimization end-to-end with machine learning. Such paradigm, called *Learn-to-Optimize* (LtO), learns a mapping between the parameters of an optimization problem and its corresponding optimal solutions using a deep neural network (DNN), as illustrated in Figure 1(c). The resulting DNN mapping is then treated as an *optimization proxy* whose role is to repeatedly solve difficult, but related optimization problems in real time (Vesselinova et al., 2020; Fioretto et al., 2020). Several LtO methods specialize in training proxies to solve difficult problem forms, especially those involving nonconvex optimization.

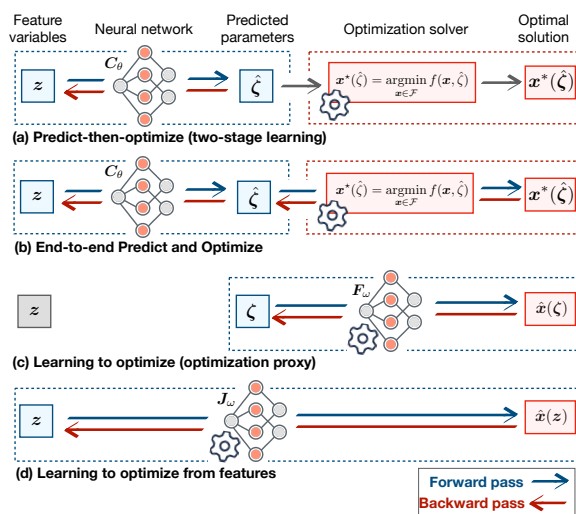

Figure 1: Illustration of Learning to Optimize from Features, in relation to other learning paradigms.

The proposed methodology of this paper, called *Learning to Optimize from Features* (LtOF), recognizes that existing Learn-to-Optimize methods can provide an array of implementations for producing learned optimization proxies, which can handle hard optimization problem forms, have fast execution speeds, and are differentiable by construction. As such, they can be adapted to the Predict-Then-Optimize setting, offering an alternative to hard optimization solvers with handcrafted backpropagation rules. However, directly transferring a pretrained optimization proxy into the training loop of an EPO model leads to poor accuracy, as shown in Section 3, due to the inability of LtO proxies to generalize outside their training distribution. To circumvent this distributional shift issue, this paper shows how to adapt the LtO methodology to learn optimal solutions directly from features.

**Contributions.** In summary, this paper makes the following novel contributions: **(1)** It investigates the use of pretrained LtO proxy models as a means to approximate the decision-making component of the PtO pipeline, and demonstrates a distributional shift effect between prediction and optimization models that leads to loss of accuracy in end-to-end training. **(2)** It proposes Learning to Optimize from Features (LtOF), in which existing LtO methods are adapted to learn solutions to optimization problems directly from observable features, circumventing the distribution shift effect over the problem parameters. **(3)** The generic LtOF framework is evaluated by adapting several well-known LtO methods to solve Predict-then-Optimize problems with difficult optimization components, under complex feature-to-parameter mappings. Besides the performance improvement over two-stage approaches, *the results show that difficult nonconvex optimization components can be incorporated into PtO pipelines naturally*, extending the flexibility and expressivity of PtO models.

## 2 PROBLEM SETTING AND BACKGROUND

In the Predict-then-Optimize (PtO) setting, a (DNN) prediction model $C_\theta : \mathcal{Z} \to \mathcal{C} \subseteq \mathbb{R}^k$ first takes as input a feature vector $z \in \mathcal{Z}$ to produce predictions $\hat{\zeta} = C_\theta(z)$. The model $C$ is itself parametrized by learnable weights $\theta$. The predictions $\hat{\zeta}$ are used to parametrize an optimization model of the form (1), which is then solved to produce optimal decisions $x^\star(\hat{\zeta}) \in \mathcal{X}$. We call these two components, respectively, the *first* and *second* stage models. Combined, their goal is to produce decisions $x^\star(\hat{\zeta})$ which minimize the ground-truth objective value $f(x^\star(\hat{\zeta}), \zeta)$ given an observation of $z \in \mathcal{Z}$. Concretely, assuming a dataset of samples $(z, \zeta)$ drawn from a joint distribution $\Omega$, the goal is to learn a model $C_\theta : \mathcal{Z} \to \mathcal{C}$ producing predictions $\hat{\zeta} = C_\theta(z)$ which achieves

$$\underset{\theta}{\text{Minimize}} \ \mathbb{E}_{(z,\zeta)\sim\Omega} \left[ f\left( x^\star(\hat{\zeta}), \zeta \right) \right]. \tag{2}$$

This optimization is equivalent to minimizing expected *regret*, defined as the magnitude of suboptimality of $\boldsymbol{x}^\star(\hat{\zeta})$ with respect to the ground-truth parameters:

$$regret(\boldsymbol{x}^\star(\hat{\zeta}), \zeta) = f(\boldsymbol{x}^\star(\hat{\zeta}), \zeta) - f(\boldsymbol{x}^\star(\zeta), \zeta). \qquad (3)$$

**Two-stage Method.** A common approach to training the prediction model $\hat{\zeta} = \boldsymbol{C}_\theta(\boldsymbol{z})$ is the *two-stage* method, which trains to minimize the mean squared error loss $\ell(\hat{\zeta}, \zeta) = \|\hat{\zeta} - \zeta\|_2^2$, without taking into account the second stage optimization. While directly minimizing the prediction errors is confluent with the task of optimizing ground-truth objective $f(\boldsymbol{x}^\star(\hat{\zeta}), \zeta)$, the separation of the two stages in training leads to error propagation with respect to the optimality of downstream decisions, due to misalignment of the training loss with the true objective (Elmachtoub & Grigas, 2021).

**End-to-End Predict-Then-Optimize.** Improving on the two-stage method, the End-to-end Predict-end-Optimize (EPO) approach trains directly to optimize the objective $f(\boldsymbol{x}^\star(\hat{\zeta}), \zeta)$ by gradient descent, which is enabled by finding or approximating the derivatives through $\boldsymbol{x}^\star(\hat{\zeta})$. This allows for end-to-end training of the PtO goal (2) directly as a loss function, which consistently outperforms two-stage methods with respect to the evaluation metric (2), especially when the mapping $\boldsymbol{z} \to \zeta$ is *difficult to learn* and subject to significant prediction error. Such an integrated training of prediction and optimization is referred to as *Smart Predict-Then-Optimize* (Elmachtoub & Grigas, 2021), *Decision-Focused Learning* (Wilder et al., 2019), or End-to-End Predict-Then-Optimize (EPO) (Tang & Khalil, 2022). This paper adopts the latter term throughout, for consistency. Various implementations of this idea have shown significant gains in downstream decision quality over the conventional two-stage method. See Figure 1 (a) and (b) for an illustrative comparison, where the constraint set is denoted with $\mathcal{F}$. An overview of related work on the topic is reported in Appendix A.

CHALLENGES IN END-TO-END PREDICT-THEN-OPTIMIZE

Despite their advantages over the two-stage, EPO methods face two key challenges: **(1) Differentiability**: the need for handcrafted backpropagation rules through $\boldsymbol{x}^\star(\zeta)$, which are highly dependent on the form of problem (1), and rely on the assumption of derivatives $\frac{\partial \boldsymbol{x}^\star}{\partial \zeta}$ which may not exist or provide useful descent directions, and require that the mapping (1) is unique, producing a well-defined function; **(2) Efficiency**: the need to solve the optimization (1) to produce $\boldsymbol{x}^\star(\zeta)$ for each sample, at each iteration of training, which is often inefficient even for simple optimization problems.

This paper is motivated by a need to address these disadvantages. To do so, it recognizes a body of work on training DNNs as *learned optimization proxies* which have fast execution, are automatically differentiable by design, and specialize in learning mappings $\zeta \to \boldsymbol{x}^\star(\zeta)$ of hard optimization problems. While the next section discusses why the direct application of learned proxies as differentiable optimization solvers in an EPO approach tends to fail, Section 4 presents a successful adaptation of the approach in which optimal solutions are learned end-to-end from the observable features $\boldsymbol{z}$.

## 3 EPO WITH OPTIMIZATION PROXIES

The Learning-to-Optimize problem setting encompasses a variety of distinct methodologies with the common goal of learning to solve optimization problems. This section characterizes that setting, before proceeding to describe an adaptation of LtO methods to the Predict-Then-Optimize setting.

**Learning to Optimize.** The idea of training DNN models to emulate optimization solvers is referred to as *Learning-to-Optimize (LtO)* (Kotary et al., 2021b). Here the goal is to learn a mapping $\boldsymbol{F}_\omega : \mathcal{C} \to \mathcal{X}$ from the parameters $\zeta$ of an optimization problem (1) to its corresponding optimal solution $\boldsymbol{x}^\star(\zeta)$ (see Figure 1 (c)). The resulting *proxy* optimization model has as its learnable component a DNN denoted $\hat{\boldsymbol{F}}_\omega$, which may be augmented with further operations $\boldsymbol{S}$ such as constraint corrections or unrolled solver steps, so that $\boldsymbol{F}_\omega = \boldsymbol{S} \circ \hat{\boldsymbol{F}}_\omega$. While training such a lightweight model to emulate optimization solvers is in general difficult, it is made tractable by restricting the task over a *limited distribution* $\Omega^F$ of problem parameters $\zeta$.

A variety of LtO methods have been proposed, many of which specialize in learning to solve problems of a specific form. Some are based on supervised learning, in which case precomputed solutions $\boldsymbol{x}^\star(\zeta)$ are required as target data in addition to parameters $\zeta$ for each sample. Others are *self-supervised*,

requiring only knowledge of the problem form (1) along with instances of the parameters $\boldsymbol{\zeta}$ for supervision in training. LtO methods employ special learning objectives to train the proxy model $\boldsymbol{F}_\omega$:

$$\underset{\omega}{\text{Minimize}} \ \mathbb{E}_{\boldsymbol{\zeta} \sim \Omega^F} \left[ \ell^{\text{LtO}} \Big( \boldsymbol{F}_\omega(\boldsymbol{\zeta}), \boldsymbol{\zeta} \Big) \right], \tag{4}$$

where $\ell^{\text{LtO}}$ represents a loss that is specific to the LtO method employed. A primary challenge in LtO is ensuring the satisfaction of constraints $\boldsymbol{g}(\hat{\boldsymbol{x}}) \leq 0$ and $\boldsymbol{h}(\hat{\boldsymbol{x}}) = 0$ by the solutions $\hat{\boldsymbol{x}}$ of the proxy model $\boldsymbol{F}_\omega$. This can be achieved, exactly or approximately, by a variety of methods, for example iteratively retraining Equation (4) while applying dual optimization steps to a Lagrangian loss function (Fioretto et al., 2020; Park & Van Hentenryck, 2023), or designing $\mathcal{S}$ to restore feasibility (Donti et al., 2021), as reviewed in Appendix B. In cases where small constraint violations remain in the solutions $\hat{\mathbf{x}}$ at inference time, they can be removed by post-processing with efficient projection or correction methods as deemed suitable for the particular application (Kotary et al., 2021b).

EPO WITH PRETRAINED OPTIMIZATION PROXIES

Viewed from the Predict-then-Optimize lens, learned optimization proxies have two beneficial features by design: **(1)** they enable very fast solving times compared to conventional solvers, and **(2)** are differentiable by virtue of being trained end-to-end. Thus, a natural question is whether it is possible to use a pre-trained optimization proxy to substitute the differentiable optimization component of an EPO pipeline. Such an approach modifies the EPO objective (2) as:

$$\underset{\theta}{\text{Minimize}} \ \mathbb{E}_{(\boldsymbol{z}, \boldsymbol{\zeta}) \sim \Omega} \left[ f \Big( \overbrace{\boldsymbol{F}_\omega \underbrace{\big( \boldsymbol{C}_\theta(\boldsymbol{z}) \big)}_{\hat{\boldsymbol{\zeta}}}}^{\hat{\boldsymbol{x}}}, \boldsymbol{\zeta} \Big) \right], \tag{5}$$

in which the solver output $\boldsymbol{x}^\star(\hat{\boldsymbol{\zeta}})$ of problem (2) is replaced with the prediction $\hat{\boldsymbol{x}}$ obtained by LtO model $\boldsymbol{F}_\omega$ on input $\hat{\boldsymbol{\zeta}}$ (gray color highlights that the model is pretrained, before freezing its weights $\omega$).

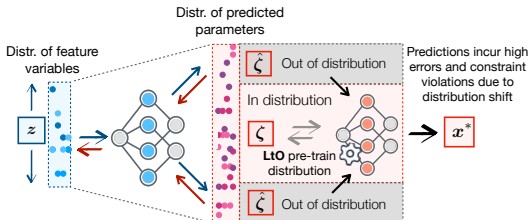

Figure 2: A distribution shift between the training distribution of a LtO proxy and the parameter predictions during training leads to inaccuracies in the proxy solver.

However, a fundamental challenge in LtO lies in the inherent limitation that ML models act as reliable optimization proxies *only within the distribution of inputs they are trained on*. This challenges the implementation of the idea of using pretrained LtOs as components of an end-to-end Predict-Then-Optimize model as the weights $\theta$ update during training, leading to continuously evolving inputs $\boldsymbol{C}_\theta(\boldsymbol{z})$ to the pretrained optimizer $\boldsymbol{F}_\omega$. Thus, to ensure robust performance, $\boldsymbol{F}_\omega$ must generalize well across virtually any input during training. However, due to the dynamic nature of $\theta$, there is an inevitable *distribution shift* in the inputs to $\boldsymbol{F}_\omega$, destabilizing the EPO training.

Figures 2 and 3 illustrate this issue. The former highlights how the input distribution to a pretrained proxy drifts during EPO training, adversely affecting both output and backpropagation. The latter quantifies this behavior, exemplified on a simple two-dimensional problem (described in Appendix C), showing rapid increase in proxy regret as $\hat{\boldsymbol{\zeta}}$ diverges from the initial training distribution $\boldsymbol{\zeta} \sim \Omega^F$ (shown in black). The experimental results presented later in Table 2 reinforce these observations. While each proxy solver performs well within its training distribution, their effectiveness deteriorates sharply when utilized as described in equation 5. This degradation is observed irrespective of any normalization applied to the proxy's input parameters during EPO training.

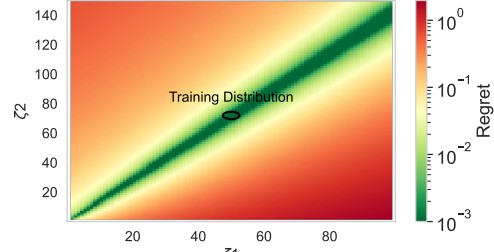

Figure 3: Effect on regret as LtO proxy acts outside its training distribution.

A step toward resolving this distribution shift issue allows the weights of $\boldsymbol{F}_\omega$ to adapt to its changing inputs, by *jointly* training the prediction and optimization models:

$$\underset{\theta, \omega}{\text{Minimize}} \ \mathbb{E}_{(\boldsymbol{z}, \boldsymbol{\zeta}) \sim \Omega} \left[ f \Big( \overbrace{\boldsymbol{F}_\omega \underbrace{\big( \boldsymbol{C}_\theta(\boldsymbol{z}) \big)}_{\hat{\boldsymbol{\zeta}}}}^{\hat{\boldsymbol{x}}}, \boldsymbol{\zeta} \Big) \right]. \tag{6}$$

The predictive model $C_\theta$ is then effectively absorbed into the predictive component of $F_\omega$, resulting in a *joint* prediction and optimization proxy model $J_\phi = F_\omega \circ C_\theta$, where $\phi = (\omega, \theta)$. Given the requirement for feasible solutions, the training objective (6) must be replaced with an LtO procedure that enforces the constraints on its outputs. This leads us to the framework presented next.

## 4 LEARNING TO OPTIMIZE FROM FEATURES

The distribution shift effect described above arises due to the disconnect in training between the first-stage prediction network $C_\theta : \mathcal{Z} \to \mathcal{C}$ and the second-stage optimization proxy $F_\omega : \mathcal{C} \to \mathcal{X}$. However, the Predict-Then-Optimize setting (see Section 2) ultimately only requires the combined model to produce a candidate optimal solution $\hat{x} \in \mathcal{X}$ given an observation of features $z \in \mathcal{Z}$. Thus, the intermediate prediction $\hat{\zeta} = C_\theta(z)$ in Equation (6) is, in principle, not needed. This motivates the choice to learn direct mappings from features to optimal solutions of the second-stage decision problem. The joint model $J_\phi : \mathcal{Z} \to \mathcal{X}$ is trained by Learning-to-Optimize procedures, employing

$$\underset{\phi}{\text{Minimize}} \ \mathbb{E}_{(z,\zeta)\sim\Omega}\left[\ell^{\text{LtO}}\Big(J_\phi(z), \zeta\Big)\right]. \tag{7}$$

This method can be seen as a direct adaptation of the Learn-to-Optimize framework to the Predict-then-Optimize setting. The key difference from the typical LtO setting, described in Section 3, is that problem parameters $\zeta \in \mathcal{C}$ are not known as inputs to the model, but the correlated features $z \in \mathcal{Z}$ are known instead. Therefore, estimated optimal solutions now take the form $\hat{x} = J_\phi(z)$ rather than $\hat{x} = F_\omega(\zeta)$. Notably, this causes the self-supervised LtO methods to become *supervised*, since the ground-truth parameters $\zeta \in \mathcal{C}$ now act only as target data while the separate feature variable $z$ takes the role of input data.

We refer to this approach as *Learning to Optimize from Features (LtOF)*. Figure 1 illustrates the key distinctions of LtOF relative to the other learning paradigms studied in the paper. Figures (1c) and (1d) distinguish LtO from LtoF by a change in model's input space, from $\zeta \in \mathcal{C}$ to $z \in \mathcal{Z}$. This brings the framework into the same problem setting as that of the two-stage and end-to-end PtO approaches, illustrated in Figures (1a) and (1b). The key difference from the PtO approaches is that they produce an estimated optimal solution $x^\star(\hat{\zeta})$ by using a true optimization solver, but applied to an imperfect parametric prediction $\hat{\zeta} = C_\theta(z)$. In contrast, LtOF directly estimates optimal solution $\hat{x}(z) = J_\phi(z)$ from features $z$, circumventing the need to represent an estimate of $\zeta$.

### 4.1 SOURCES OF ERROR

Both the PtO and LtOF methods yield solutions subject to *regret*, which measures suboptimality relative to the true parameters $\zeta$, as defined in Equation 3. However, while in end-to-end and, especially, in the two-stage PtO approaches, the regret in $x^\star(\hat{\zeta})$ arises from imprecise parameter predictions $\hat{\zeta} = C_\theta(z)$ (Mandi et al., 2023), in LtOF, the regret in the inferred solutions $\hat{x}(z) = J_\phi(z)$ arises due to imperfect learning of the proxy optimization. This error is inherent to the LtO methodology used to train the joint prediction and optimization model $J_\phi$, and persists even in typical LtO, in which $\zeta$ are precisely known. In principle, a secondary source of error can arise from imperfect learning of the implicit feature-to-parameter mapping $z \to \zeta$ within the joint model $J_\phi$. However, these two sources of error are indistinguishable, as the prediction and optimization steps are learned jointly. Finally, depending on the specific LtO procedure adopted, a further source of error arises when small violations to the constraints occur in $\hat{x}(z)$. In such cases, restoring feasibility (e.g, through projection or heuristics steps) often induces slight increases in regret (Fioretto et al., 2020).

Despite being prone to optimization error, Section 5 shows that Learning to Optimize from Features greatly outperforms two-stage methods, and is competitive with EPO training based on exact differentiation through $x^\star(\zeta)$, when the feature-to-parameter mapping $z \to \zeta$ is complex. This is achieved *without* any access to exact optimization solvers, nor models of their derivatives. This feat can be explained by the fact that by learning optimal solutions end-to-end directly from features, LtOF does not directly depend on learning an accurate representation of the underlying mapping from $z$ to $\zeta$.

### 4.2 EFFICIENCY BENEFITS

Because the primary goal of the Learn-to-Optimize methodology is to achieve *fast solving times*, the LtOF approach broadly inherits this advantage. While these benefits in speed may be diminished when constraint violations are present and complex feasibility restoration are required, efficient feasibility restoration is possible for many classes of optimization models Beck (2017). This enables the design of *accelerated* PtO models within the LtOF framework, as shown in Section 5.

### 4.3 MODELING BENEFITS

While EPO approaches require the implementation of problem-specific backpropagation rules, the LtOF framework allows for the utilization of existing LtO methodologies in the PtO setting, on a problem-specific basis. A variety of existing LtO methods specialize in learning to solve convex and nonconvex optimization (Fioretto et al., 2020; Park & Van Hentenryck, 2023; Donti et al., 2021), combinatorial optimization (Bello et al., 2017; Kool et al., 2019), and other more specialized problem forms (Wu & Lisser, 2022). The experiments of this paper focus on the scope of continuous optimization problems, whose LtO approaches share a common set of solution strategies.

## 5 EXPERIMENTAL RESULTS

The LtOF approach is evaluated on three Predict-Then-Optimize tasks, each with a distinct second stage optimization component $x^\star : \mathcal{C} \to \mathcal{X}$, as in equation 1. These tasks include a convex quadratic program (QP), a nonconvex quadratic programming variant, and a nonconvex program with sinusoidal constraints, to showcase the general applicability of the framework.

**LtOF methods.** Three different LtOF implementations are evaluated on each applicable PtO task, based on distinct Learn-to-Optimize methods, reviewed in detail in Appendix B. These include:

- *Lagrangian Dual Learning (**LD**)* (Fioretto et al., 2020), which augments a regression loss with penalty terms, updated to mimic a Lagrangian Dual ascent method to encourage the satisfaction of the problem's constraints.
- *Self-supervised Primal Dual Learning (**PDL**)* (Park & Van Hentenryck, 2023), which uses an augmented Lagrangian loss function to perform joint self-supervised training of primal and dual networks for solution estimation.
- *Deep Constraint Completion and Correction (**DC3**)* (Donti et al., 2021), which relies on a completion technique to enforce constraint satisfaction in predicted solutions, while maximizing the empirical objective function in self-supervised training.

While several other Learn-to-Optimize methods have been proposed in the literature, the above-described collection represents diverse subset which is used to demonstrate the potential of adapting the LtO methodology as a whole to the Predict-Then-Optimize setting.

**Feature generation**. End-to-End Predict-Then-Optimize methods integrate learning and optimization to minimize the propagation of prediction errors–specifically, from feature mappings $z \to \zeta$ to the resulting decisions $x^\star(\zeta)$ (regret). It's crucial to recognize that *even methods with high error propagation* can yield low regret *if the prediction errors are low*. To account for this, EPO studies often employ synthetically generated feature mappings to control prediction task difficulty (Elmachtoub & Grigas, 2021; Mandi et al., 2023). Accordingly, for each experiment, we generate feature datasets $(z_1, \dots z_N) \in \mathcal{Z}$ from ground-truth parameter sets $(\zeta_1, \dots \zeta_N) \in \mathcal{C}$ using random mappings of increasing complexity. A feedforward neural network, $G^k$, initialized uniformly at random with $k$ layers, serves as the feature generator $z = G^k(\zeta)$. Evaluation is then carried out for each PtO task on feature datasets generated with $k \in \{1, 2, 4, 8\}$, keeping target parameters $\zeta$ constant.

**Baselines**. In our experiments, LtOF models use feedforward networks with $k$ hidden layers. For comparison, we also evaluate two-stage and, where applicable, EPO models, using architectures with $k$ hidden layers where $k \in \{1, 2, 4, 8\}$. Further training specifics are provided in Appendix D.

**Comparison to LtO setting**. It is natural to ask how solution quality varies when transitioning from LtO to LtOF in a PtO setting, where solutions are learned directly from features. To address this question, each PtO experiment includes results from its analogous Learning to Optimize setting, where a DNN $\mathbf{F}_\omega : \mathcal{C} \to \mathcal{X}$ learns a mapping from the parameters $\zeta$ of an optimization problem to its

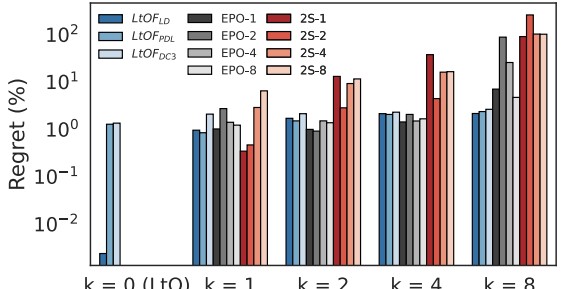

| | Method | Portfolio | N/conv. QP | AC-OPF |
|---|---|---|---|---|
| **LtOF** | LD it | **0.0003** | 0.0000 | **0.0004** |
| | LD fct | 0.0000 | 0.0045 | 0.1573 |
| | PDL it | **0.0003** | 0.0000 | 0.0006 |
| | PDL fct | 0.0000 | 0.0045 | 0.1513 |
| | DC3 it | 0.0011 | 0.0001 | - |
| | DC3 fct | 0.0003 | 0.0000 | - |
| **PtO** | PtO-1 et | 0.0054 | 0.0122 | 0.1729 |
| | PtO-2 et | 0.0059 | 0.0104 | 0.1645 |
| | PtO-4 et | 0.0062 | 0.0123 | 0.1777 |
| | PtO-8 et | 0.0067 | 0.0133 | 0.1651 |

Figure 4: Comparison between **LtO** ($k=0$), **LtOF**, Two-stage (**2S**) and **EPO** ($k > 1$) on the portfolio optimization. 2S(EPO)-$m$ indicates that the prediction model of the respective PtO method is an $m$ layer ReLU neural network.

Table 1: Execution (*et*), inference (*it*), and feasibility correction (*fct*) times for **LtOF** and **PtO** (in seconds) for each sample. **Two-stage** methods execution times are comparable to PtO's ones.

corresponding solution $x^\star(\zeta)$. This is denoted $k=0$ (LtO), indicating the absence of any feature mapping. All figures report the regret obtained by LtO methods for reference, although they are not directly comparable to the Predict-then-Optimize setting.

Finally, all reported results are averages across 20 random seeds and we refer the reader to Appendix D for additional details regarding experimental settings, architectures, and hyperparamaters adopted.

## 5.1 CONVEX QUADRATIC OPTIMIZATION

A well-known problem combining prediction and optimization is the Markowitz Portfolio Optimization (Rubinstein, 2002). This task has as its optimization component a convex Quadratic Program:

$$x^\star(\zeta) = \arg\max_{x \geq 0} \ \zeta^T x - \lambda x^T \Sigma x, \quad \text{s.t.} \ \mathbf{1}^T x = 1 \tag{8}$$

in which parameters $\zeta \in \mathbb{R}^D$ represent future asset prices, and decisions $x \in \mathbb{R}^D$ represent their fractional allocations within a portfolio. The objective is to maximize a balance of risk, as measured by the quadratic form covariance matrix $\Sigma$, and total return $\zeta^T x$. Historical prices of $D = 50$ assets are obtained from the Nasdaq online database (Nasdaq, 2022) and used to form price vectors $\zeta_i$, $1 \leq i \leq N$, with $N = 12,000$ individual samples collected from 2015-2019. In the outputs $\hat{x}$ of each LtOF method, possible feasibility violations are restored, at *low computational cost*, by first clipping $[\hat{x}]_+$ to satisfy $x \geq 0$, then dividing by its sum to satisfy $\mathbf{1}^T x = 1$. The convex solver cvxpy (Diamond & Boyd, 2016) is used as the optimization component in each PtO method.

**Results**. Figure 4 shows the percentage regret due to LtOF implementations based on *LD*, *PDL* and *DC3*. Two-stage and EPO models are evaluated for comparison, with predictive components given various numbers of layers. For feature complexity $k > 1$, each LtOF model outperforms the best two-stage model, increasingly with $k$ and up to nearly *two orders of magnitude* when $k = 8$. The EPO model, trained using exact derivatives through (8) as provided by the differentiable solver in cvxpylayers (Agrawal et al., 2019a) is competitive with LtOF until $k = 4$, after which point its best variant is outperformed by each LtOF variant. This result showcases the ability of LtOF models to reach high accuracy under complex feature mappings *without* access to optimization problem solvers *or* their derivatives, in training or inference, in contrast to conventional PtO frameworks.

Table 1 presents LtOF inference times (*it*) and feasibility correction times (*fct*), which are compared with the per-sample execution times (*et*) for PtO methods. Run times for two-stage methods are closely aligned with those of EPO, and thus obmitted. Notice how the LtOF methods are at least an order of magnitude faster than PtO methods. This efficiency has two key implications: firstly, the per-sample speedup can significantly accelerate training for PtO problems. Secondly, it is especially advantageous during inference, particularly if data-driven decisions are needed in real-time.

## 5.2 NONCONVEX QP VARIANT

As a step in difficulty beyond convex QPs, this experiment considers a generic QP problem augmented with an additional oscillating objective term, resulting in a *nonconvex* optimization component:

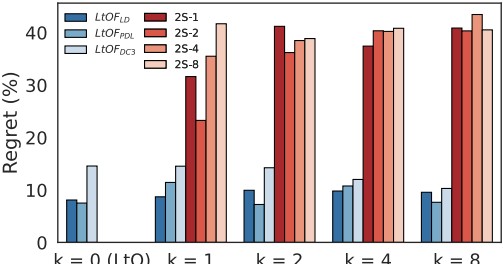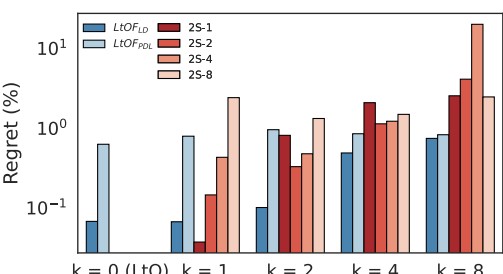

Figure 5: Comparison between LtO ($k = 0$), LtOF, and Two Stage Method (2S) on the nonconvex QP (**left**) and AC-OPF case (**right**). Right plot y-axis is in log-scale.

$$\mathbf{x}^\star(\boldsymbol{\zeta}) = \arg\min_{\boldsymbol{x}} \ \frac{1}{2}\boldsymbol{x}^T\boldsymbol{Q}\boldsymbol{x} + \boldsymbol{\zeta}^T\sin(\boldsymbol{x})$$
$$\texttt{s.t.} \ \ \boldsymbol{Ax} = \boldsymbol{b}, \ \boldsymbol{Gx} \le \boldsymbol{h},$$

in which the $\sin$ function is applied elementwise. This formulation was used to evaluate the LtO methods proposed both in Donti et al. (2021) and in Park & Van Hentenryck (2023). Following those works, $\boldsymbol{0} \preccurlyeq \boldsymbol{Q} \in \mathbb{R}^{n \times n}$, $\boldsymbol{A} \in \mathbb{R}^{n_{\text{eq}} \times n}$, $\boldsymbol{b} \in \mathbb{R}^{n_{\text{eq}}}$, $\boldsymbol{G} \in \mathbb{R}^{n_{\text{ineq}} \times n}$ and $\boldsymbol{h} \in \mathbb{R}^{n_{\text{ineq}}}$ have elements drawn uniformly at random. Here it is evaluated as part of a Predict-Then-Optimize pipeline in which predicted coefficients occupy the nonconvex term. Feasibility is restored by a projection onto the feasible set, which is calculated by a more efficiently solvable *convex* QP. The problem dimensions are $n = 50$ $n_{\text{eq}} = 25$, and $n_{\text{ineq}} = 25$.

**Results.** Figure 5 (left) shows regret due to LtOF models based on *LD*, *PDL* and *DC3*, along with two-stage baseline PtO methods. No EPO baselines are available due to the optimization component's nonconvexity. The best two-stage models perform poorly for most values of $k$, implying that the regret is particularly sensitive to prediction errors in the oscillating term. Thus its increasing trend with $k$ is less pronounced than in other experiments. The best LtOF models achieve over $4\times$ times lower regret than the best baselines, suggesting strong potential for this approach in contexts which require predicting parameters of non-linear objective functions. Additionally, the fastest LtOF method achieves up to three order magnitude speedup over the two-stage, after restoring feasibility.

## 5.3 NONCONVEX AC-OPTIMAL POWER FLOW

Given a vector of marginal costs $\boldsymbol{\zeta}$ for each power generator in an electrical grid, the AC-Optimal Power Flow problem optimizes the generation and dispatch of electrical power from generators to nodes with predefined demands. The objective is to minimize cost, while meeting demand exactly. The full optimization problem and more details are specified in Appendix C, where a quadratic cost objective is minimized subject to nonconvex physical and engineering power systems constraints. This experiment simulates a energy market situation in which generation costs are as-yet unknown to the power system planners, and must be estimated based on correlated data. The overall goal is to predict costs so as to minimize cost-regret over an example network with 54 generators, 99 demand loads, and 118 buses taken from the well-known NESTA energy system test case archive (Coffrin et al., 2014). Feasibility is restored for each LtOF model by a projection onto the nonconvex feasible set. Optimal solutions to the AC-OPF problem, along with such projections, are obtained using state-of-the-art Interior Point OPTimizer IPOPT (Wächter & Laird, 2023).

**Results.** Figure 5 (right) presents regret percentages, comparing LtOF to a two-stage baseline. Note that no general EPO exists for handling such nonconvex decision components. *DC3* is also omitted, following Park & Van Hentenryck (2023), due to its incompatibility with the LtO variant of this experiment. Notice how the *best* two-stage model is outperformed by the (supervised) *LD* variant of LtOF for $k > 1$, and also by the (self-supervised) *PDL* variant for $k > 2$. Notably, PDL appears robust to increases in the feature mapping complexity $k$. On the other hand, it is outperformed in each case by the LD variant. Notice how, in the complex feature mapping regime $k > 1$, the best LtOF variant achieves up to an order of magnitude improvement in regret relative to the most competitive

| | Method | Portfolio | | | Nonconvex QP | | | AC-OPF | | |
|---|---|---|---|---|---|---|---|---|---|---|
| | | $k=2$ | $k=4$ | $k=8$ | $k=2$ | $k=4$ | $k=8$ | $k=2$ | $k=4$ | $k=8$ |
| LtOF | LD Regret | 1.7170 | 2.1540 | **2.1700** | 9.9279 | **9.7879** | 9.5473 | **0.1016** | **0.4904** | **0.7470** |
| | LD Regret (*) | 1.5739 | 2.0903 | 2.1386 | 9.9250 | 9.8211 | 9.5556 | 0.0013 | 0.0071 | 0.0195 |
| | LD Violation (*) | 0.0010 | 0.0091 | 0.0044 | 0.0148 | 0.0162 | 0.0195 | 0.0020 | 0.0037 | 0.0042 |
| | PDL Regret | 1.5150 | 2.0720 | 2.3830 | **7.2699** | 10.747 | **7.6399** | 0.9603 | 0.8543 | 0.8304 |
| | PDL Regret (*) | 1.4123 | 1.9372 | 2.0435 | 7.2735 | 10.749 | 7.6394 | 0.0260 | 0.0243 | 0.0242 |
| | PDL Violation (*) | 0.0001 | 0.0003 | 0.0003 | 0.0028 | 0.0013 | 0.0015 | 0.0000 | 0.0002 | 0.0002 |
| | DC3 Regret | 2.1490 | 2.3140 | 2.6600 | 14.271 | 11.028 | 10.666 | - | - | - |
| | DC3 Regret (*) | 2.0512 | 1.9584 | 2.3465 | 13.779 | 11.755 | 10.849 | - | - | - |
| | DC3 Violation (*) | 0.0000 | 0.0000 | 0.0000 | 0.5158 | 0.5113 | 0.5192 | - | - | - |
| EPO | EPO Regret (Best) | **0.9220** | **1.4393** | 4.7495 | - | - | - | - | - | - |
| | EPO w/ Proxy Regret (Best) | 154.40 | 119.31 | 114.69 | 812.75 | 804.26 | 789.50 | 389.04 | 413.89 | 404.74 |
| | Two-Stage Regret (Best) | 2.8590 | 4.4790 | 91.326 | 36.168 | 37.399 | 38.297 | 0.3300 | 1.1380 | 2.4740 |

Table 2: Percentage Regret and Constraint Violations for all experiments. (*) denotes "Before Restoration". Missing entries denote experiments not achievable with the associated method.

two-stage method. Despite feasibility of the LtOF methods being restored by a (difficult) projection onto nonconvex constraint set, modest speed advantages are nonetheless gained.

Table 2 collects an abridged set of accuracy results due to each LtOF implementation and PtO baseline, across all experimental tasks. Complete results can be found in Appendix D. In particular, average constraint violations and objective regret are shown in addition to regret after restoring feasibility. Infeasible results are shown in grey for purposes of comparing the regret loss due to restoration. Best results on each task are shown in bold. Additionally, regret achieved by the EPO framework with pretrained proxies (discussed in Section 3) are included. Average regrets between 100 and 800 percent illustrate the effect of their distributional shifts on accuracy. Again, notice how, in the context of complex feature mappings $k \geq 2$, LtOF is competitive with EPO, while bringing substantial computational advantages, and consistently outperforms two-stage methods, often, beyond an order of magnitude in regret.

## 6 LIMITATIONS, DISCUSSION, AND CONCLUSIONS

The primary *advantage* of the Learning to Optimize from Features approach to PtO settings is its generic framework, which enables it to leverage a variety of existing techniques and methods from the LtO literature. On the other hand, as such, a particular implementation of LtOF may inherit any limitations of the specific LtO method that it adopts. For example, when the LtO method does not ensure feasibility, the ability to restore feasibility may be need as part of a PtO pipeline. Future work should focus on understanding to what extent a broader variety of LtO methods can be applied to PtO settings; given the large variety of existing works in the area, such a task is beyond the scope of this paper. In particular, this paper does not investigate of the use of *combinatorial* optimization proxies in learning to optimize from features. Such methods tend to use a distinct set of approaches from those studied in this paper, including problem-specific designs which assume special structure in the optimization problem, and often rely on training by reinforcement learning (Bello et al., 2017; Kool et al., 2019; Mao et al., 2019). As such, this direction is left to future work.

The main *disadvantage* inherent to any LtOF implementation, compared to end-to-end PtO, is the inability to recover parameter estimations from the predictive model, since optimal solutions are predicted end-to-end from features. Although it is not required in the canonical PtO problem setting, this may present a complication in situations where transferring the parameter estimations to external solvers is desirable. This presents an interesting direction for future work.

By showing that effective Predict-Then-Optimize models can be composed purely of Learning-to-Optimize methods, this paper has aimed to provide a unifying perspective on these as-yet distinct problem settings. The flexibility of its approach has been demonstrated by showing superior performance over PtO baselines with diverse problem forms. As the advantages of LtO are often best realized in combination with application-specific techniques, it is hoped that future work can build on these findings to maximize the practical benefits offered by Learning to Optimize in settings that require data-driven decision-making.

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
