# OpenReview forum: "Predict-then-Optimize  via Learning to Optimize from Features"
_ICLR.cc/2024/Conference — ICLR 2024 Conference Withdrawn Submission_

### Official Review · Reviewer_Lqhu · 2023-10-31

**Soundness:** 1 poor
**Presentation:** 3 good
**Contribution:** 1 poor
**Rating:** 3
**Confidence:** 3

**Summary:**

The paper introduces several ways to address a predict-then-optimize problem and investigates how to adopt learning-to-optimize methods to it. Through this, the paper presents a way to address predict-then-optimize problems by using learning-to-optimize methods.

**Strengths:**

The paper has a clear presentation and is easy to follow. The figures are helpful in understanding the content, and the existing concepts in related works are introduced properly.

**Weaknesses:**

The analysis and investigation in the paper is reasonable, but seems trivial. There is no rigorous mathematical analysis on the use of learning-to-optimize methods for predict-then-optimize problems, and there are no novel algorithmic improvements or suggestions as well. In summary, the idea of the paper is simply applying existing learning-to-optimize to predict-then-optimize problems. This proposal seems too simple.

**Questions:**

1. The proposal suggests that existing learning-to-optimize methods can be applied to predict-then-optimize problems. This idea (i.e., the application of learning-to-optimize for predict-then-optimize problems) seems simple to think of. What is the contribution of the paper?
2. In a similar context, is there any further discussion? For example, a variant of learning-to-optimize for the idea or a tailored loss function for effective learning. It seems hard to claim the idea is a new framework based on the current paper.
3. The authors state that the distribution shift issue is caused by a pretrained optimization proxy. Thus, a proper solution to the issue should be a way to resolve the distribution shift issue while using a pretrained proxy. In this regard, simply training the prediction part and learning-to-optimize part jointly, as the paper suggests, does not seem to be a solution to that problem, because it means that pretrained proxies cannot be used.
4. The proposed approach involves prediction due to its predict-then-optimize nature. So, it would be difficult to generate a pretrained model and utilize it for different predict-then-optimize problems. Then, even if the proposed approach can achieve the short inference time once the learning-to-optimize model is trained, it suffers from the training cost because of getting various $(z,x^*)$ samples in the process of collecting datasets for training.

---

> ### Author Response · Authors · 2023-11-14
> **Rebuttal**
>
> We thank the reviewer for their review.
>
> **This idea seems simple to think of. What is the contribution of the paper?**
>
> The contribution of the paper is to show that by adapting Learning to Optimize methods to the Predict-Then-Optimize domain, we introduce an entirely new approach to PtO, which leads to high-accuracy solutions while retaining the benefits of LtO including increased speed and adaptability to various problem forms.
>
> The fact that this can be done efficiently and without access to the typical toolset including optimization solvers or their derivatives, has not been demonstrated before and is of interest to many in the community. We are glad that the paper's presentation makes the idea seem "simple to think of".  The reviewer should have to explain why this should be grounds for rejecting the paper. The work is novel.
>
> **A proper solution to the distribution shift issue would use a pretrained proxy.**
>
> We do not understand this critique. What would be the point of using a pretrained proxy, if the distributional shift issue can be circumvented by jointly training the proxy?
>
>
> **There is no rigorous mathematical analysis.**
>
> A rigorous mathematical analysis would not be appropriate for the contribution of this paper. It describes a generic framework, and any particular analysis would depend on the LtO training method used. As such we support the paper's proposed framework by experimentally evaluating over several distinct LtO methods to show its general effectiveness. The LtO methods, in general, are also rarely supported by rigorous proofs and related papers tend not to focus on this aspect. We are not trying to make a contribution in the LtO domain.
>
>
> **"It suffers from the training cost because of getting various training samples..."**
>
> This is not true. It's important for the reviewers to note this: two out of three of our LtOF implementations (DC3 and PDL) are completely self-supervised and do not require precomputation of optimal solutions for their training sets. Please also see our response to Reviewer Q2Xf as it pertains to training set generation in their cited works [1],[2],and [3]. Those works all rely on precomputation of optimal solutions and require generating a plurality of surrogate training sets to support the overall training process. Our proposal is capable of avoiding this pitfall by self-supervised end-to-end learning of the prediction and proxy optimization components. We hope that understanding our contribution within this context can help the reviewer see why it is interesting and relevant.
>
>
> **Is there any further discussion? Any new variant of LtO for this?**
>
> The effectiveness of the framework in solving PtO problems is ultimately what matters. A "new variant" of Learning-to-Optimize is not needed to achieve this; we demonstrate the effectiveness of adapting several already-existing variants. This is the point of the paper: a whole new class of methods can be adapted to this PtO problem setting. Such suggestions seem to be focusing more on the appearance of the contribution being too simple, rather than the novelty and effectiveness of the contribution on the important problem which it solves.

---

> > ### Comment · Reviewer_Lqhu · 2023-11-15
> >
> > Here is the response to the rebuttal:
> >
> > I disagree that the authors have introduced an entirely new approach to PtO. 1) LtO is an approach that already existed, 2) the proposed approach is essentially just pipelining LtO into a prediction model (although the detailed implementation may differ), and 3) the way that LtO is used in the proposed approach is hardly novel. Therefore, for this paper to have a sufficient contribution, we need either a rigorous analysis to show that the proposed approach is effective or new ideas to maximize the effectiveness of the proposed approach. However, no such thing can be found in the paper.
> >
> > The authors stated the distribution shift issue when using pretrained proxies, but the proposed approach simply avoids the distribution shift issue by training the prediction part and LtO together without considering pretrained proxies. As a result, LtOF may be free from the distribution shift issue. However, this is a way to solve the issue at the expense of learning the optimization proxy from the scratch. (I got that two of the three implemented LtOs in the experiments do not require the samples, $(z,x^*)$, but anyway, that comment was in the context of the expense of additional learning.) Roughly speaking, if we could allow the expense for training an optimization proxy, training the proxy for the distribution of $\hat\zeta$ might free it from the distribution shift issue even not using LtOF.
> >
> > Because the rebuttal does not address my concerns about the paper's lack of contributions, I am keeping my score.

---

> ### Author Response · Authors · 2023-11-16
>
> The contribution has been made clear. Predict-Then-Optimize learning has primarily been based on modeling derivatives through optimization problems or the associated regret functions. By identifying a way to adapt existing LtO methods to this problem setting, the paper shows a way to accomplish the same task with high performance without using such derivatives at all. This change in perspective, among other advantages, brings with it a natural way to handle complex problems with nonconvex constraints in PtO, which was previously challenging.
>
> Responding to:
>
> "The way that LtO is used in the proposed approach is hardly novel.": This is a factual claim that's not appropriate in an open review without providing references to support it. Why not provide a reference to show where it has already been done?
>
> "Training the proxy for the distribution of $\hat{\zeta}$ might free it from the distribution shift issue even not using LtOF.":  This assumes the existence of a known distribution over $\hat{\zeta}$ on which to train. Finding a model which maps distribution $z$ to  $\hat{\zeta}$ in a way that minimizes regret is the PtO problem itself. If this exists then the problem is already solved.
>
> The main critique of this review seems to be that the paper is not novel because it combines existing frameworks in a way that "seems simple". But combining them in a new way, leading to new practical advantages, is by definition novel. This manner of critique is only possible in the absence of any discussion of the proposal's demonstrated advantages.

---

### Official Review · Reviewer_rTig · 2023-10-31

**Soundness:** 2 fair
**Presentation:** 3 good
**Contribution:** 2 fair
**Rating:** 3
**Confidence:** 4

**Summary:**

This paper studies the predict-then-optimize framework where an optimization problem with part of the parameters missing and a feature correlated to the missing parameters are given. The conventional way of solving predict-then-optimize problems is to learn a machine learning model to map from features to missing parameters based on a training dataset. Then the model can be used to predict missing parameters from unseen features to solve optimization problems accordingly. This solution is known as two-stage learning.

Recently, an end-to-end predict-then-optimize learning framework, e.g., smart predict-then-optimize, decision-focused learning, decision-aware learning, was proposed to learn the machine learning model end-to-end with optimization problems in the training loop. The idea is to make optimization differentiable so that the solution quality and regret can be used to backpropagate gradients to learn the machine learning models. This end-to-end predict-then-optimize (EPO) framework has achieved many successes in different applications, but also suffering from its limitation on differentiability and computation cost.

The authors instead propose to generalize from the idea of learning to optimize (LtO) to achieve end-to-end learning. The idea of learning to optimize is based on learning a machine learning model to solve an optimization problem. This resolves the limitation of differentiability of optimization problems because machine learning models such as neural networks are differentiable by construction. This approach also reduces the computation cost of backpropagation because backpropagation in neural networks is efficient. The authors propose to jointly learn the learning to optimize framework and the prediction problem to achieve end-to-end learning in the predict-then-optimize problem. This idea leads to their solution “learning to optimize from features (LtOF)”.

Lastly, the authors run experiments on three different domains with convex and non-convex objectives and different loss functions to compare their LtOF solution with LtO, EPO, and two-stage learning. Their results show that LtOF can achieve better performance than two-stage learning while EPO sometimes can be better but suffer from limitation of differentiability and computation cost.

**Strengths:**

The authors clearly specify the challenges in end-to-end predict-then-optimize framework and propose solutions to handle them. The proposed method is also generic and can be used in many applications.

**Weaknesses:**

**Robustness**: I don’t think the LtOF approach can handle the issue of distribution shift that is encountered by LtO. When a different distribution of features and optimization parameters is present, the authors do not address why this approach is more robust to distribution shift. In addition, LtOF can be impacted by distribution shift more because it learns both optimal solution and parameter predictive model jointly, while EPO assumes given optimization problem and only learns the parameter predictive model. Given the same amount of training data, LtOF needs to learn more parameters and I believe this will lead to poorer generalization unless you have sufficient data.

**Training v.s. testing sets**: I didn’t see any explanations on whether the experimental results shown in the paper are from the training set or from the testing set. If it is from the training set, then the results are not surprised given that LtOF jointly learns how to solve optimization and the predictive model. LtOF has more parameters and thus can overfit to the training set more than other methods. LtOF also uses the regret or decision quality as the training objective. So it is not surprised that LtOF outperforms two-stage learning. Please clarify if testing set is used and how you verify generalizability.

[updated] I noticed in the appendix that you briefly mention it. Please emphasize it in the main paper and clarify how the test sets are constructed.

**Nonconvex EPO**: In fact, EPO can handle nonconvex optimization objectives by locally approximating nonconvex objectives around the local minimum as convex objectives due to local optimality. The (heuristic) differentiability of nonconvex objectives was also mentioned in Amos et al. 2017, Agrawal et al. 2019a, Perrault et al. 2020, and Wang et al. 2020. The authors should also compare with EPO on problems with nonconvex objectives. The experimental results shown in the paper show that EPO can indeed be better in portfolio optimization in Table 2, but the authors claim that EPO is not available for nonconvex tasks. I think adding EPO results for nonconvex versions would help justify the authors’ claim.

**Synthetic dataset**: all the datasets used in the paper are synthetic with generated features. It is not clear if the proposed method would work in real-world applications and still outperform other existing methods.



**References for nonconvex EPO**:
- Perrault, Andrew, et al. "End-to-end game-focused learning of adversary behavior in security games." Proceedings of the AAAI Conference on Artificial Intelligence. 2020.
- Wang, Kai, et al. "Scalable Game-Focused Learning of Adversary Models: Data-to-Decisions in Network Security Games." AAMAS. 2020.

**Questions:**

Please refer to the weaknesses section.

---

> ### Author Response · Authors · 2023-11-14
> **Rebuttal**
>
> We thank the reviewer for their review.
>
> **"I don’t think the LtOF approach can handle the issue of distribution shift."**
>
> The reviewer may have missed the main point in our description of the distribution shift, which specifically refers to a divergence between the training distribution of problem parameters for the LtO proxy, and the inputs encountered  by the proxy as parametric predictions during subsequent training of the EPO model. We are not concerned, in this work, about a shift in the distribution of input features to the final prediction model, as studied e.g. in domain adaptation.
>
> Recall that the distributional shift is not measured directly, but rather by its effect on the accuracy of the downstream solutions (regret). What ultimately matters in PtO is the regret incurred by downstream decisions. The fact that our LtOF implementations soundly outperform the two-stage baseline models, and are competitive with EPO models, shows in itself that the distributional shift issue has been effectively handled. The concept behind this is to close the loop between optimization proxy training and EPO prediction training. Since there are no longer two separate training processes, there can be no distributional shift between them to speak of.
>
>
> **"In fact, EPO can handle nonconvex optimization objectives by locally approximating..."**
>
> We appreciate the references provided on this point. We agree that it may be the case that some form of convex EPO adaptation could be constructed to provide another baseline for the nonconvex experiments. However, there is some subtlety and we disagree with the assertion that nonconvex problems in general admit useful convex approximations for EPO training.
>
> For example the first cited work provided by the reviewer (Perrault) uses a convex QP approximation to the nonconvex model at its optimal point as an EPO surrogate. This technique is situational, and does not apply in general. If the objective Hessian is not positive semidefinite, the QP approximation will not be convex. As an example, consider PtO with a nonconvex QP as its optimization component; its Hessian is constant and non-Positive Semidefinite. So it does not admit a convex QP approximation anywhere. Also, it seems clear that our AC-OPF problem, with nonconvex constraints and linear objective, would not produce a useful differentiable QP surrogate. For one, it would be an LP (the Hessian is zero), which is known to be non-differentiable wrt the predicted objective coefficients. The citations to Amos 2017 and Agrawal 2019 also do not provide any evidence that a convex surrogate technique should work in general for nonconvex PtO.
>
> While reasonable EPO alternatives have been contrived for some nonconvex models, it has not been systematically shown that there is a reliable method for doing so in general. So we disagree with the assumption that such techniques obviously exist, and need to be evaluated as baselines.
>
> **"Synthetic Datasets"**
>
>
> We incorporated real-world data from Nasdaq to populate the parameter dataset $\zeta$ of the first experiment, but we  make clear in the paper that the underlying features $z$ are all generated synthetically. This is because synthetic data allows us to make a stronger evaluation by controlling the complexity of the feature mapping from $z$ to $\zeta$. By showing that LtOF outperforms the twostage methods to an increasing degree as the feature complexity increases, and is competitive with EPO where available, we can give stronger evidence that LtOF has an advantage in mitigating error propagation from parametric predictions to downstream decisions. The original "Smart Predict Then Optimize" of Elmachtoub 2017 uses synthetic data for both features and parameters, for the same reason. In contrast, several works use the "Warcraft Shortest Path" benchmark to make a  visually appealing baseline, but in practice the ResNet feature extractor leads to very good two-stage performance, which does not allow for the advantage of the EPO methods to be fully expressed.

---

> > ### Comment · Reviewer_rTig · 2023-11-16
> > **Thank you for your clarification**
> >
> > I want to thank the authors for the detailed response. I also hope some of my feedback is helpful for the authors.
> >
> > **Distribution shift**
> > Thank you for clarifying it. I do see the difference. Indeed the distribution shift I meant is the common shift in the distribution of input features.
> >
> > **Local approximation**
> > Local approximation can be non-convex but still backpropagatable using qpth or cvxpylayer as long as the solution is a local minimum in the constrained optimization problem. Or alternatively (equivalently), you can use the same KKT conditions trick on non-convex objectives (see discussion section in the cvxpylayer paper by Agrawal 2019 and its reference to Gould et al. 2020).
> >
> > As the responses from the authors suggested, there is a disagreement in whether this kind of non-convex differentiable optimization-based methods should be considered as a reliable and systematical baseline. IMO, technical-wise, it is well-known (or at least published since 2019-2020) and applicable to your setting (non-convex). So it is undoubtedly a reasonable baseline. This part I can't compromise given that the authors' main contribution is methodological and experimental, and claiming that LtOF is better than EPO in these domains.
> >
> > **Synthetic datasets**
> > I totally understand the reasons why you want to use synthetic datasets. This is not a negative point but also doesn't add value to my evaluation though.
> >
> >
> > Overall, I keep my assessment. But I think some part of the paper is certainly interesting but just needs more justification and analysis. For example, distribution shift between the output predictions and the LtO input parameters is interesting. How this is addressed by your end-to-end method (beyond just using end-to-end learning) and what guarantees or any theoretical properties does your method show, unfortunately, are not shown in the paper though.
> >
> > Again, I thank the authors for their responses and hope my response clarifies my assessment.

---

> ### Author Response · Authors · 2023-11-16
>
> We thank the reviewer for their thoughtful and balanced feedback on the paper. The reviewers' suggestions about local approximation seem reasonable and may help inform future improvements to the paper.
>
> Responding to one point:
>
> Differentiating the KKT points in nonconvex optimization does sound like a reasonable suggestion to pursue for a baseline result, in principle. At the same time, several factors make it difficult to test, including that this method would be predicated on finding the globally optimal solution to a nonconvex problem, for each sample at each iteration of training, which makes the practicality of the experiment questionable. Together with the fact that such an approach hasn't been shown to work reliably in general, to the point that it is well-accepted as an effective solution for nonconvex PtO (it could reasonably doubted; gradients don't always coincide with descent directions in the nonconvex case), we think it's not quite appropriate that such a baseline be required.

---

### Official Review · Reviewer_BPgy · 2023-10-31

**Soundness:** 3 good
**Presentation:** 3 good
**Contribution:** 2 fair
**Rating:** 5
**Confidence:** 3

**Summary:**

This paper studies the predict-then-optimize framework. Predict-then-optimize framework is where some parameters of an optimization problem are unknown and need to be predicted based on some features. There are mainly two types of previous work mentioned in this paper. One is a two-stage approach which first predicts the parameters as a supervised learning problem and then solves the optimization problem with the predicted parameters. The other type is End-to-End Predict-Then-Optimize (EPO) which gets the gradient through the optimization problem with respect to the input features. The two-stage approach does not consider the downstream optimization problem in the supervised learning task as the loss function cannot match the regret of the final optimization problem. This regret represents the gaps of the objective of the optimization problem with predicted parameters and true parameters.

Existing work uses deep learning methods to predict the solution to an optimization problem, i.e., a mapping from the parameters of optimization problem to optimal decisions. This paper applies some methods of training proxy models into the Predict-then-Optimize framework. The advantage of using this model in PtO is that the NN representation of the optimization problem is differentiable and fast compared to solving the optimization problem itself. However, a naive proxy model suffers poor performance if predicting on unseen data. A proxy model can predict some decision which does not satisfy the constraints and a naive projection to feasible region approach may not have good performance. This paper uses three recent works of learning proxy model, Lagrangian Dual Learning (LD), Self-Supervised Primal-Dual Learning (PDL), and Deep Constraint Completion and Correction (DC3) to address the generalization and constraints violations issues. They apply these proxy models techniques to the predict-then-optimize framework to get the Learning to Optimize from Features (LtOF). They paper evaluate LtOF with the previous two-stage and EPO approaches.

**Strengths:**

1. The use of LtO techniques in the PtO setting and the simultaneous training of the LtO and the PtO model is novel.
2. LtOF method performs better than the EPO and two-stage approaches in the nonconvex QP problem and nonconvex AC-optimal power flow problems and is very fast.
3. There are substantial technical difficulties involved with the problems they select due to their difficult constraint regions. Not having an explicit optimization solve makes getting feasible solutions difficult, but their models do pretty well on this score.
4. The paper is clearly written.

**Weaknesses:**

1. The contribution is kind of slight. There really isn't much to do to get LtOF—or the way the paper is written makes it seem like this combination is relatively straightforward. It gives the impression that, if there is a difficulty, is in the empirical details around hyperparameters and getting the LtO to satisfy the constraints, but the authors do not claim a contribution in this area.
2. The experiments don't seem to compare to strong EPO methods (or in problems where EPO is at least better than two-stage). For two of the three problems, the EPO comparison is worse than 2-stage. This leaves the impression that experimental validation does not fully evaluate how strong LtO is relative to EPO (it comes across more that it is stronger than two-stage in cases where EPO cannot be applied).
3. It would be useful to see the dependence on the amount of training instances available AND to have info about the amount of training time that is required (if I haven't missed them, these timing results are not present—it seems that a major cost of PtOF is the training cost.)

**Questions:**

1. Hope this paper could detail more about some techniques used in LtoF methods to reduce the negative effect of constraints violations. The Appendix C and figure 7 describe a projection method to find a feasible solution based on an approximate optimization solution for the AC Load Flow model. One of the LtOF methods uses the Deep Constraint Completion and Correction (DC3). Is the projection method only used in Lagrangian Dual Learning (LD) and Self-Supervised Primal-Dual Learning (PDL) LtOF but not DC3 LtOF? Is it possible that the projection method always needs to hand-craft a specific function for proxy models of different optimization problems? Or can learning proxy models of very different optimization problems can utilize a general projection method for the constraints violation issue?

---

> ### Author Response · Authors · 2023-11-14
> **Rebuttal**
>
> We appreciate that the reviewer points out the difficulty of the task which requires satisfaction of difficult, nonconvex constraints, and the novelty of the paper's solution which accomplishes this without access to optimization solvers or their derivatives.
>
> **"The contribution is kind of slight / the paper makes it seem straightforward."**
>
> The main idea is indeed straightforward in a way, and the paper is written to emphasize the important but widely unrecognized connection between Predict-Then-Optimize and Learning-to-Optimize. This insight, like many things, is only "obvious" once it has been explained and demonstrated. Ours is the first to demonstrate this connection, and evaluate the practical performance and potential advantages of adapting LtO in this way. Many in the community would be interested to see this connection illustrated and systematically evaluated.
>
>
>
> **"The EPO baselines do not seem strong."**
>
> The EPO baselines are correctly implemented. The hyperparameter tuning is described in the Appendix D. Figure 4 shows that each EPO baseline outperforms twostage method, for each feature mapping complexity $k$.  The one exception is that best EPO is indeed outperformed by the best two-stage on k=1, but this is normal and shows the evaluations are correctly implemented. EPO approaches are only expected to outperform when the feature mapping is sufficiently complex. Otherwise the parametric predictions are so easy to learn that the two-stage can reach almost zero error. For k=1, the feature mapping is linear.
>
>
> **"More detail about the LtO methods and constraint violations."**
>
> First, please notice the extended Related Work section in Appendix B, which gives more details about the LtO methods which we adapted in the paper. Also, it's important to remember that not all LtO methods are prone to constraint violations, so the issue does not affect LtOF in the most general case. However, the methods we evaluate in the paper are indeed prone to small constraint violations. The fact that projections must be hand-crafted in such situations is indeed a good point. On the other hand, projections are typically not nearly as difficult to implement as full optimization solvers or their derivatives. Also note that feasibility corrections are not always required; these methods allow for some parameter tuning, which can be used to reduce violations to zero, generally at some cost to optimality. For example, for LD and PDL the lagrange multiplier stepsizes can generally be increased past a threshold where constraint violations disappear. Also notice that in two of three experiments, DC3 incurs no violations in the first place and does not require a correction. On the other hand its regret post-correction is generally not as low as that of the other LtO methods.

---

### Official Review · Reviewer_Q2Xf · 2023-11-08

**Soundness:** 2 fair
**Presentation:** 3 good
**Contribution:** 2 fair
**Rating:** 3
**Confidence:** 4

**Summary:**

This paper proposes a way to extend "Learning to Optimize" (LtO) approaches to the predict-then-optimize framework. Broadly, it combines the prediction $z \to \hat{\zeta}$ and optimization $\hat{\zeta} \to x^\star$ steps (using LtO) and proposes Learning to Optimize from Features (LtOF). It evaluates LtOF against 2-stage on three domains (and against EPO for one domain) and shows improved performance and, in one case, lower computational cost.

**Strengths:**

- **The paper is mostly well-written.** I found it easy to follow the main arguments.

**Weaknesses:**

- **The paper is missing important related work.** The main motivation for this method seems to be that it's hard to differentiate through the optimization problem for arbitrary optimization problems. However, there has been work in the predict-then-optimize literature that addresses this problem exactly. Specifically, Shah et al. [1] learn ``decision losses'' that learn the mapping from $\hat{\zeta} \to regret(x^\star(\hat{\zeta}), \zeta)$ (as defined in eq. (3)) to get around the issue of having to learn differentiable surrogates (there have also been more recent follow-ups [2,3] that make this more efficient). Even more generally, because you do not consider combinatorial optimization problems but rather continuous optimization problems, you could just use numerical differentiation methods to find the exact derivatives $\frac{\partial regret}{\partial \hat{\zeta}}$. For reasonable values of $|z| = 30, 50$ (as in this paper) along with some clever optimization tricks (e.g., warm-starting, GPU-acceleration, etc.) this may not be doable. There has also been a paper [4] that says you can get away with very simple surrogate gradients as well. All in all, these are just the tip of the iceberg. This space is not nearly as unexplored as the related work in this paper suggests, and it is important to compare LtOF to these alternate approaches.

- **The experiments are poorly designed and implemented.** I have several concerns with the experiments in this paper:
  1. *Two out of three domains do not have reasonable baselines:* Nonconvex QP and AC-OPF compare only to two-stage and not any EPO alternatives. As this paper has noted, the fact that you can do better than two-stage is well-known at this point. The question is about whether you can do better than other reasonable EPO alternatives like [1].
  1. *No comparison to prediction + pre-trained LtO:* One simple baseline would be to compare to optimizing over a pre-trained LtO model. In Figures 2 and 3 you argue that this is bad, but never show experimental results on it? If (a) you train the LtO model on the distribution of $\zeta$ and then (b) you pre-train the predictive model on a two-stage loss and then fine-tune it on a pre-trained LtO model, the distribution shift may not be too bad and perhaps it may not perform too badly? Is this the EPO with Proxy Regret baseline? It doesn't seem to be defined anywhere. Do you do any of the things suggested above to improve performance? How do you do hyperparameter tuning? Overall, this baseline seems to be a straw man.
  1. *Other concerns:* Where are the error bars? There seems to be an overload of $k$ as a measure of complexity of the mapping of $z \to \zeta$ and also a measure of complexity for $F_\omega$. Often for EPO, it works better to have a simpler predictive model $F_\omega$ due to overfitting issues; have you tried this? Also, it is often useful to "mix a little two-stage loss" into the EPO objective. Have you tried this?

- **I don't think that the *idea* is particularly novel/interesting.** Taken together, I believe that you *could* choose to solve predict-than-optimize problems using LtOF, and even that you could do better than two-stage. However, as the paper has noted as well, this has been studied time and again. Moreover, the idea of using LtO to solve for $x^\star(\zeta)$ is not particularly novel, imo. I think the more interesting question is about whether you *should* do it in this way and when, and I don't think the paper has answered this question. Specifically:
  1. *Generalization of LtOF to similar problems:* What happens if we want to change the number of stocks in our portfolio optimization problem at test time? In EPO, predictions are typically made per-stock $i$ using covariates $z_i \to \hat{\zeta}_i$. As a result, we can, for example, remove some of the stocks from our portfolio optimization problem at test-time without having to re-train.
  1. *Being able to use the optimal solver at test time:* From Table 2, we can see that the regret before restoration for the AC-OPF domain is *very* different than after restoration. This suggests that LtOF is not able to learn how to make good decisions. As a result, it seems like it would be useful to be able to use an optimal planner at test-time. This seems even more true in, e.g., shortest path or ranking problems where there is a lot of structure in the constraints. This begs the question, when should you (a) predict the intermediate parameters, e.g., using prediction + LtO separately, versus (b) predict decisions directly from features?
  1. *Interpretability/Fairness/Additional Considerations:* This is hard enough for prediction alone, does LtOF make this easier or harder in the predict-then-optimize case?

References:
1. Shah, Sanket, et al. "Decision-Focused Learning without Differentiable Optimization: Learning Locally Optimized Decision Losses." NeurIPS (2022).
2. Zharmagambetov, Arman, et al. "Landscape surrogate: Learning decision losses for mathematical optimization under partial information." arXiv preprint arXiv:2307.08964 (2023).
3. Shah, Sanket, et al. "Leaving the Nest: Going Beyond Local Loss Functions for Predict-Then-Optimize." arXiv preprint arXiv:2305.16830 (2023).
4. Sahoo, Subham Sekhar, et al. "Backpropagation through combinatorial algorithms: Identity with projection works." ICLR (2023).

**Questions:**

Please respond to the questions in the weaknesses section.

---

> ### Author Response · Authors · 2023-11-14
> **Rebuttal**
>
> We appreciate the reviewer's references to related works [1],[2], and [3].  We respond to their main critiques below.
>
> **"The paper is not novel/interesting in light of related works (cited)."**
>
> We first point out that existence of recent work in the area shows that the community is indeed interested in this research direction (PtO with optimization surrogates). Alternative approaches such as ours should also be accommodated, especially since each of the existing works cited by the reviewer has its own drawbacks, which we will partially summarize below.
>
> Reference [4] cited by the reviewer is not based on surrogate learning, and is not comparable to our paper in its aims or approach. Of the remaining three works [1], [2], [3] advocated by the reviewer, [3] is not yet published, and [2] was just announced as accepted in NeurIPS. The fact that these works are yet unpublished or brand new, means that our paper should be considered as concurrent work. We will cite these works in the updated article, but to state that our paper is not novel or interesting in light of these works, and advocate rejection on that basis, is unfair.
>
> It's also important to point out the differences in our proposal from the goals shared by [1],[2],[3]. Those works focus on learning surrogate loss functions for use in EPO training, in which the end product is a predictor of $\zeta$. Ours learns a joint prediction and optimization model. Our learned model functions like an LtO proxy, and outputs full optimal solutions directly. The main thing shared in common among ours and [1],[2],[3] seems to be the goal of training PtO without optimization in-the-loop.
>
> It is important for all to note that our paper has relevance in light of the drawbacks of [3]. That paper tries to learn a surrogate loss function for PtO, which is then used in subsequent EPO training, free of optimization. Since that learned surrogate loss function is only valid on its training distributions, the method of [3] is also subject to the Distributional Shift effect discussed in our paper. As described in Section 6 of its ArXiv version, the paper [3] copes with the distributional shift effect by running many, differently parametrized mock training routines in parallel, from which predictions are sampled throughout training, and saved to later train the global surrogate loss function. Since there is no way to ensure that the resulting training set will produce a good surrogate loss, the paper [3] relies on a brute force method of guessing and testing surrogate losses trained by data sourced from the various mock training runs, which is waved away as a "hyperparameter" search, until good end task performance is found. The published paper [1], which is a prequel to [3], proposes to cope with the distributional shift by requiring to train a different "local surrogate loss" for each sample of each training set. Both are inefficient designs especially given that they require precomputation of optimal solutions over each training set made in the overall process  (as detailed in Section 5 Paragraph 1 of [3]).
>
> Our paper is particularly relevant in light of these drawbacks, because it explicitly demonstrates the distributional shift issue between surrogate training and EPO training, and tackles this problem directly by closing the loop between optimization surrogate and EPO prediction models using end-to-end learning. This allows the method to avoid resorting to "fine-tuning" the entire training set as suggested by the reviewer.  The reviewer suggests that failing to pursue such an approach is a drawback of our paper, but it is not at all clear that any principled approach to surrogate-based EPO can be built upon such a concept. We use optimization rather than hyperparameter search to deal with the distributional shift, and the self-supervised variants shown in our paper are efficiently trainable without precomputation of training sets of optimal solutions. This is an important advantage which the reviewer does not recognize.
>
> The works [1],[2],[3] may have good merit within their scope and may each offer certain advantages. There is room in the literature for all these different approaches. But the reviewer holds that those approaches are considered "novel/interesting" while ours is not.  The reviewer should clarify the choice to reject on those grounds.
>
> [1/2]

---

> > ### Comment · Reviewer_Q2Xf · 2023-11-14
> > **Factual Clarification about Publication Status of Cited Papers**
> >
> > I will respond to the authors' second set of comments separately, but I first wanted to make an important clarification. Decision-Focused Learning without Differentiable Optimization [1] was published at NeurIPS-22 ([link to proceedings](https://proceedings.neurips.cc/paper_files/paper/2022/hash/0904c7edde20d7134a77fc7f9cd86ea2-Abstract-Conference.html)) and Landscape Surrogate [2] was *accepted*, not rejected, to NeurIPS-23 ([list of accepted papers](https://neurips.cc/virtual/2023/poster/70329)). In addition, while [2, 3] can be considered concurrent work, given that [1] was first on Arxiv on *30 Mar 2022* and published in NeurIPS-22, I believe it constitutes prior work that should be compared to.

---

> > > ### Author Response · Authors · 2023-11-15
> > >
> > > Thanks for the correction. Our claim that [3] was not accepted was made in error, due to misinterpreting NeurIPS website terminology. DBLP and Google Scholar have not yet been updated with word of acceptance. We have edited our response to reflect this.
> > >
> > > This does not affect the main points of our response, whether it is brand newly accepted or still under review.

---

> ### Author Response · Authors · 2023-11-14
>
> In particular, could you please respond to the following questions:
>
> - Would you agree that the works [1],[2],[3] all face a very similar distributional shift issue as in our paper, and propose to mitigate the issue in different ways?
> - The solution proposed in [1] is inefficient to the degree that a follow-up paper [3] was warranted, implying that surrogate learning in PtO is still an open problem.  Is that fair to say?
> - You confidently suggest that "fine-tuning" the dataset is an obvious solution to the distributional shift issue, to the degree that our pre-trained proxy EPO is characterized as a Strawman. This is because that is the solution proposed in [3]. Is that correct?
> - We point out that the "fine-tuning" approach of [3] essentially mitigates the distributional shift effect by hyperparameterization of the entire data generation process of the surrogate's training routine. Meaning that entire training sets must be generated for the surrogate loss in a guess-and-test manner, until good performance on the end task is found. Is it fair to say that there is still room for improvement in this aspect, for future work?
> - Your critique does not address the core contribution of the paper, which is to identify the distributional shift as the main impediment to surrogate learning for PtO, and then mitigate the issue with an end-end learning-to-optimize solution. Would you please give an explanation of why our solution is not "novel/interesting" in context of the proposed solutions in [1] and [3], given your close familiarity with those works?
>
>
>
> **"The paper does not compare against EPO with pretrained proxy."**
>
> This is not correct, please see Table 2 which lists the regret incurred by EPO training with pretrained proxy, also referenced in the distributional shift discussion of Section 3 Page 4. As for the suggestion that the distributional shift can be alleviated by "fine tuning" the training set: we tried several such things which did not work, including iteratively retraining the proxy with out-of-distribution samples encountered in EPO training. There is no reliable way to  anticipate the parametric predictions encountered during EPO training and use them to pretrain the LtO proxy in advance. Cited work [3] attempts something similar and resorts to inefficient "hyperparameterization" of the entire dataset generation to get good results as explained above.  There is no "straw man" here.
>
> **"Two out of three experiments do not use reasonable baselines."**
>
> The reviewer refers to the experiments in  which we use nonconvex optimization components. First we point out that two-stage models are universally considered to be standard baselines, especially in the absence of a well-established EPO alternative for nonconvex problems. It is not always easy to beat the two-stage method, and doing so systematically on an array of increasingly complex datasets demonstrates that our method greatly improves error generalization from predictions to downstream decisions. This is the main goal in PtO. Please also see our response to Reviewer rTig with regards to nonconvex EPO alternatives. For problems with nonconvex constraints, there is no well-established EPO alternative such that new papers should be rejected for failing to test against it.
>
> **"This suggests that LtOF is not able to learn how to make good decisions."**
>
> The reviewer should substantiate this critique. The end regret of the solutions after feasibility restoration is substantially less than one percent in each case, and as low as $0.1$ percent. Why does this prompt the assertion that the model cannot learn good decisions?
>
>
> **"What about interpretability, fairness, etc."**
>
> These are tangential issues that distract from a discussion about our paper's core contributions. Same can be said of the remark on optimal planning at test time. These aspects are irrelevant with respect to the canonical PtO problem setting which we are studying. Will the reviewer address any flaws with the core contribution of the paper, specifically our solution to the distributional shift problem in relation the works they cited, or the actual performance of our methods?
>
> [2/2]

---

> > ### Comment · Reviewer_Q2Xf · 2023-11-15
> > **Response to Rebuttal**
> >
> > I'll respond point-wise:
> > 1. **Prior Work vs. Concurrent Work**: As I've highlighted separately, [1] has been published at NeurIPS-22 and [2] was accepted to NeurIPS-23. Given that [1] was published nearly a year ago and its first version was put on Arxiv in March 2022, I believe that it constitutes *prior work* and **not** concurrent work.
> > 1. **Addressing Distribution Shift**: I do agree that [1, 2, 3] had to address a distribution shift issue that is similar to this paper. I also agree that [1, 3] address these in different ways from this paper. However, (a) [2] does something similar to this paper by interleaving the predictive model training with the surrogate loss training, but more importantly (b) the fact that these alternate ways to deal with distribution shift seem to work well in [1, 3] raises the question of whether LtOF is strictly necessary to use LtO in the PtO context. The reason that I called the experiments "strawman" (I now realize that this is pejorative and that I shouldn't have used it; I apologize) is because I believe you did not give LtO a "fair shot", i.e., use some straightforward heuristics to try to mitigate the distribution shift issue and, in failing, motivate the need for LtOF. You say that ``we tried several such things which did not work``, but there is no documentation of these. Also, the baseline "EPO with Proxy Regret" is not defined anywhere (including the appendix), so I wasn't sure if (a) this was indeed LtO, or even if it was (b) what data trained LtO on.
> > 1. **Baselines for non-linear EPO**: In your response to reviewer rTig, you conclude with `While reasonable EPO alternatives have been contrived for some nonconvex models, it has not been systematically shown that there is a reliable method for doing so in general.` I believe that your paper would be significantly stronger if you show that this is indeed true in your experiments by comparing it to one of these proposed alternatives. As it stands, it's not clear where LtOF stands in comparison to *any* of the past approaches in the literature (contrived or not). This is also where I believe that [4] comes in, it shows that even seemingly ridiculous surrogates can have surprisingly good performance.
> > 1. **LtOF for AC-OPF**: I'm sorry, I didn't realize it was percentage regret. I thought it was fractional (between 0 and 1). You are correct.
> >
> > All-in-all, my takeaway from the paper as it stands currently is that (a) LtOF is one possible approach to PtO with non-convex optimization problems, and (b) that it works better than two-stage for such problems. However, the negatives are that (a) it doesn't compare to past work in the EPO literature that addresses similar concerns (e.g., [1] or the baselines rTig suggests), and (b) there are no ablations that describe how important the key contribution of using LtOF to mitigate distribution shift is (cf. LtO + heuristics to mitigate distribution shift). Given this, I keep my original score.

---

> ### Author Response · Authors · 2023-11-15
>
> We'll give our response point-wise as well:
>
> 1. Benchmarking Against LODLs ([1]): We make no claim that [1] is concurrent work, it is surely prior. Our main point is that in light of its own admitted drawbacks, and since it is not directly comparable to LtOF in its goals (we learn optimizers and not loss functions, the advantages are different) makes it unreasonable to reject a paper based on failure to benchmark against it. Even [3], which has the same first author as [1], critiques the inefficient approach of [1], which requires the user to train entirely separate local surrogate loss functions for $\textbf{each sample}$ of the master training set. While [1] may have practical merit, it is not a stretch to call this a brute-force method. Computational resources are expensive, and this method [1] cannot be a required baseline for all surrogate learning proposals going forward.
>
> 2. "We tried several things that did not work": In the response above it's mentioned that we specifically tried iterative retraining of the proxy based on out-of-distribution samples collected in prior training. It didn't yield any positive result, so there is no clear reason to document this in the paper. There is also no clear conceptual reason why it should work in the first place, so inclusion of such a thing in our paper would be out of place. Yet the reviewer states failure to include such results as a main reason for maintaining their score (see point (b) above).
>
> 3. The fact that we demonstrate a method that beats the only accepted baseline (two-stage) on nonconvex problems is therefore clearly valuable as a contribution of our paper. No valid convex differentiable approximation of the AC-OPF problem has yet been suggested in this review, yet failure to benchmark against it is still used to justify rejection.
>
> 4. It's ok, thank you for the correction.
>
> Note that the reviewer still insists on the baselines suggested by rTig, but we have explained in our response to rTig why this cannot work on our AC-OPF example. Their suggested convex QP approximation is not even differentiable in our case (it has zero Hessian).
>
> The reviewer concludes above that the main drawbacks of the paper are failure to benchmark against nonconvex surrogates, LODLs ([1]), and an unspecified "LtO + heuristics to mitigate distribution shift". We believe that our response here shows that none of those critiques are reasonable. We have directly refuted the existence of a differentiable convex QP surrogate to evaluate as a baseline for nonconvex AC-OPF, and it's not even specified what "heuristics" are being asked for. The issue with LODLs is addressed above. To maintain a score of 3 on such grounds does seems unjustified.
>
> It seems clear that we have incorporated an effective solution to the distributional shift issue, which is interesting and novel in context of the reviewer's cited works, and yields practical advantages, but this fact is still not really acknowledged. It's also not refuted.

---

### Author Response · Authors · 2023-11-14
**Summary of response**

We thank the reviewers for their feedback. Several of the provided references are valuable for the discussion; we have taken the time to study all the provided references.

A variety of critiques have been brought forth. We find most of the technical critiques easy to address and have done so in the individual responses. However it seems that the committee's view of the paper's main weaknesses are of a more subjective nature. We paraphrase each such point and respond to it:

**"The paper is not novel or interesting in light of recent related work (Reviewer Q2Xf)":**

It's important for the committee to follow our first response to Q2Xf on this point. It goes into detail about why our proposal is relevant in the context of related work cited by this reviewer, that is currently also under review right now. Our paper solves the distributional shift issue in proxy-based PtO by using end-to-end learning, and the significance of this solution is much better appreciated in light of the alternative solutions proposed in the works cited [1],[2],[3] by Q2Xf. We believe that understanding this discussion will help clarify the significance of our paper, and the importance of the problem it addresses, to the other reviewers.

**"The paper leaves out important baselines (Q2Xf, rTig)":**

Multiple reviewers assume that important baselines have been left out; this is a major criticism that has not been substantiated in the review. Please see the response to rTig on the issue of failing to benchmark against convex EPO surrogate models: such method clearly cannot be viable, for example, on our AC-OPF problem with linear objective and non-convex constraints. Meanwhile Q2Xf holds that the methods must be benchmarked against [4], which simply describes a heuristic shortcut to model the jacobian of an optimization as an identity matrix. This has never been considered a standard baseline and is not well-supported to be effective in general. Can the reviewers show any published work in the PtO scope which compares with this as a baseline? Finally, we find the suggestion to require benchmarking against the cited works [1],[2],[3] to be particularly unreasonable, for reasons detailed in the response to Q2Xf. Two of those works are unpublished or brand new and uncited. They are far from considered as accepted standards in the space, and require significant computational burden (more detail below) and human effort to implement. Further, they are not directly comparable to our method because we do not try to learn surrogate loss functions for use in EPO, but rather surrogate optimization models for application to PtO.

**"The proposal seems too simple."**

Our article shows the effectiveness of our proposal in solving an important problem. The paper's end-to-end solution of the distributional shift problem is significant in the context of several other works that are actively being developed (see above). It also shows a conceptual connection between LtO and PtO problem settings which can be practically exploited leading to a new class of approaches to PtO, along with a systematical evaluation and demonstration of it. This comes, as explained in the paper, with advantages over EPO approaches with respect to runtime and flexibility over problem forms. The idea may "seem simple", after it has already been explained and demonstrated. The reviewers should consider whether this is really a weakness of the paper and grounds for rejection while overlooking the conceptual contributions and practical advantages.